# The SARS-CoV-2-Inactivating Activity of Hydroxytyrosol-Rich Aqueous Olive Pulp Extract (HIDROX^®^) and Its Use as a Virucidal Cream for Topical Application

**DOI:** 10.3390/v13020232

**Published:** 2021-02-02

**Authors:** Yohei Takeda, Dulamjav Jamsransuren, Sachiko Matsuda, Roberto Crea, Haruko Ogawa

**Affiliations:** 1Research Center for Global Agromedicine, Obihiro University of Agriculture and Veterinary Medicine, 2-11 Inada, Obihiro, Hokkaido 080-8555, Japan; ytakeda@obihiro.ac.jp; 2Department of Veterinary Medicine, Obihiro University of Agriculture and Veterinary Medicine, 2-11 Inada, Obihiro, Hokkaido 080-8555, Japan; duuya.dj@gmail.com (D.J.); chaka@obihiro.ac.jp (S.M.); 3Oliphenol LLC., 26225 Eden Landing Road, Suite C, Hayward, CA 94545, USA; robertocrea@oliphenol.com

**Keywords:** aqueous olive pulp extract, hydroxytyrosol, natural-derived material, severe acute respiratory syndrome coronavirus 2, topical application, virus inactivation

## Abstract

The severe acute respiratory syndrome coronavirus 2 (SARS-CoV-2) has spread globally. Although measures to control SARS-CoV-2, namely, vaccination, medication, and chemical disinfectants are being investigated, there is an increase in the demand for auxiliary antiviral approaches using natural compounds. Here we have focused on hydroxytyrosol (HT)-rich aqueous olive pulp extract (HIDROX^®^) and evaluated its SARS-CoV-2-inactivating activity in vitro. We showed that the HIDROX solution exhibits time- and concentration-dependent SARS-CoV-2-inactivating activities, and that HIDROX has more potent virucidal activity than pure HT. The evaluation of the mechanism of action suggested that both HIDROX and HT induced structural changes in SARS-CoV-2, which changed the molecular weight of the spike proteins. Even though the spike protein is highly glycosylated, this change was induced regardless of the glycosylation status. In addition, HIDROX or HT treatment disrupted the viral genome. Moreover, the HIDROX-containing cream applied on film showed time- and concentration-dependent SARS-CoV-2-inactivating activities. Thus, the HIDROX-containing cream can be applied topically as an antiviral hand cream. Our findings suggest that HIDROX contributes to improving SARS-CoV-2 control measures.

## 1. Introduction

Severe acute respiratory syndrome coronavirus 2 (SARS-CoV-2) emerged in China in December 2019 [1], and coronavirus disease 2019 (COVID-19) caused by this virus was declared a pandemic. Since this highly contagious virus has caused a high number of deaths [2], attempts to establish infection control measures are being made globally. More than 60 COVID-19 candidate vaccines are in the clinical trial phase and >170 are in the preclinical phase as of January 2021 [3]. Although some vaccines have been approved, vaccination of the global population will not be accomplished immediately. Many candidate drugs against SARS-CoV-2 have been discovered, and drugs such as remdesivir and corticosteroid have been reported to be therapeutically efficient in some clinical trials [4,5,6,7]. However, some studies have reported no statistically significant therapeutic benefits of these drugs [8,9,10]. The contradictory results could be because of differences in ethnicity of the tested groups, severity of symptoms, and starting time of treatment. Therefore, further studies are needed for establishing effective treatment protocols.

Demand for the development of auxiliary or complementary approaches for preventing viral infection or aggravation of symptom has increased. The use of natural compounds from plants with proven activity against pathogenic viruses is one such strategy. Exploratory research of natural components that show anti-SARS-CoV-2 activity is being actively conducted and some candidate components have been reported by us and other groups in vitro [11,12,13]. Plant polyphenols are naturally-derived compounds that show antiviral activity against various pathogenic viruses [14]. Hydroxytyrosol (HT), with a molecular weight of 154.16 g/mol, is one of the main phenolic compounds in olive (*Olea europaea* L.) extracts and olive oil [15]. Biological activities associated with olive polyphenols, including antioxidant, anti-inflammatory, anticancer, and antimicrobial [16], and antiviral (against influenza A virus (IAV) and human immunodeficiency virus (HIV)) have been reported [17,18]. Because coronavirus is a single-stranded RNA virus with an envelope, similar to IAV and HIV, we postulated that HT would have antiviral activity against SARS-CoV-2. Hence, we evaluated the SARS-CoV-2-inactivating activity of HT. In addition, the virucidal activity of standardized HT-rich aqueous olive pulp extract (HIDROX^®^ 12%; Oliphenol LLC., CA, USA), which is commercially available and produced on a large scale from the aqueous effluent of the olive oil industry, was evaluated. HIDROX^®^ 12% powder includes ~12% polyphenols, of which HT accounts for ~40% (HT accounts for ~4.8% of HIDROX). Studies have reported the safety of HIDROX in vitro and in vivo [19] and have shown its beneficial effects on health, including anti-inflammatory activity in mouse [20], protection from neurodegeneration in zebrafish [21], extension of health span in a *C. elegans* model of Parkinson’s disease [22], improvement of synovitis and inflammatory symptoms in arthritic rat [23], and improvement in activities of daily living in 99 patients with osteo- and rheumatoid arthritis [24]. Because some studies have reported that the effects of HIDROX are superior to pure HT [20,21,22], we speculated that HIDROX would inactivate SARS-CoV-2 more potently than HT. Moreover, we evaluated the applicability of HIDROX with a wide range of antiviral measures. The SARS-CoV-2-inactivating activities of HIDROX or HT were tested both in solution and in a HIDROX-containing cream, which can be applied topically or dermally.

## 2. Materials and Methods

### 2.1. Virus and Cells

SARS-CoV-2 (JPN/TY/WK-521 strain) and transmembrane protease serine 2 (TMPRSS2)-expressing VeroE6 (VeroE6/TMPRSS2) cells [25] were obtained from the National Institute of Infectious Diseases (Tokyo, Japan). For passaging, VeroE6/TMPRSS2 cells were cultured in Dulbecco’s modified Eagle’s minimal essential medium (DMEM) (Nissui Pharmaceutical Co., Ltd., Tokyo, Japan) supplemented with 10% fetal bovine serum, 2 mM l-glutamine (FUJIFILM Wako Pure Chemical Co., Osaka, Japan), 0.15% NaHCO_3_ (FUJIFILM Wako Pure Chemical Co.), 2 µg/mL amphotericin B (Bristol-Myers Squibb Co., New York, NY, USA), 100 µg/mL kanamycin (Meiji Seika Pharma Co., Ltd., Tokyo, Japan), and 100 µg/mL G418 disulfate (Nacalai Tesque Inc., Kyoto, Japan). VeroE6/TMPRSS2 cells were infected with SARS-CoV-2, cultured in viral growth medium (VGM) composed of DMEM supplemented with 1% fetal bovine serum, 20 mM l-glutamine, 0.15% NaHCO_3_, 2 µg/mL amphotericin B, and 100 µg/mL kanamycin for 3 d. Culture supernatants were stored at −80 °C as viral stock (viral titer: 7.25 log_10_ 50% tissue culture infective dose (TCID_50_)/mL). SARS-CoV-2 was handled in a biosafety level 3 facility.

### 2.2. Sample Preparation

To prepare the HIDROX solution, 1 g HIDROX^®^ 12% (Oliphenol LLC., CA, USA) powder was dissolved in 10 mL phosphate-buffer saline (PBS) and centrifuged at 850× *g* for 10 min. The aqueous layer was stored at −30 °C and used as the 100 mg/mL HIDROX stock. To prepare the HT solution, 1 g 3-hydroxytyrosol (Tokyo Chemical Industry Co., Ltd., Tokyo, Japan) was dissolved in 10 mL PBS and the 100 mg/mL HT stock was stored at −30 °C. HIDROX and HT solutions were prepared by diluting the stock solutions with PBS before use. OLIVENOL^TM^ plus^+^ Healing Moisturizer (Oliphenol LLC.) containing 2%, 5%, and 10% HIDROX was prepared and used as HIDROX-containing cream. HIDROX-free (0%) ingredients of OLIVENOL^TM^ plus^+^ Healing Moisturizer were used as the base control.

### 2.3. Evaluation of Virucidal Activity of HIDROX and HT Solutions Against SARS-CoV-2

SARS-CoV-2 solution (7.25 log_10_ TCID_50_/mL) was mixed with nine volumes of HIDROX or HT. The final concentrations of HIDROX and HT in the mixture were 0.45–11.25 and 0.05–0.90 mg/mL, respectively. As a diluent control, PBS was mixed with the viral suspension. The mixtures were incubated at 25 °C from 5 min to 24 h and then inoculated into cells, and a ten-fold serial dilution was performed. The cells were incubated for 3 d, the cytopathic effect of SARS-CoV-2 was evaluated, and viral titer (log_10_ TCID_50_/mL) was calculated using the Behrens–Kärber method [26]. The detection limit of viral titer in each test group was determined based on the cytotoxic concentration of each test solution. The detection limit was set higher in the group treated with a solution of higher cytotoxicity. PBS, 0.45 mg/mL HIDROX, and 0.05 mg/mL HT solutions did not show any cytotoxicity, and the detection limit of the viral titer in the groups treated by these solutions was set to 1.25 log_10_ TCID_50_/mL, according to our viral titer calculation. On the other hand, ≥0.90 mg/mL HIDROX and 0.90 mg/mL HT solutions demonstrated cytotoxicity. The detection limit in the groups treated with 0.90, 4.50, 5.63 mg/mL HIDROX, and 0.90 mg/mL HT solutions was set to 2.25 log_10_ TCID_50_/mL and the detection limit in the group treated by 11.25 mg/mL HIDROX solution was set to 3.25 log_10_ TCID_50_/mL, respectively.

### 2.4. Western Blotting

To evaluate the impact of the test compound on the structure of viral proteins, including spike (S) protein S1 subunit, S protein receptor-binding domain (RBD), S protein S2 subunit, and nucleocapsid (N) protein, the SARS-CoV-2 solution (7.25 log_10_ TCID_50_/mL) or each recombinant viral protein was mixed with nine volumes of HIDROX or HT. The product information for recombinant viral proteins is shown in Appendix A. The final concentration of HIDROX and HT in the mixture was 0.90 mg/mL. As a diluent control, PBS was mixed with the viral suspension or recombinant viral proteins. The final concentration of recombinant proteins in the mixture was approximately 10 µg/mL. To analyze the interaction of HIDROX or HT with carbohydrate chains expressed on S proteins, approximately 10 µg/mL of recombinant S proteins (S1 subunit, RBD, and S2 subunit) was incubated at 37 °C overnight with or without 6500 units/mL of PNGase F PRIME^TM^ Glycosidase (N-Zyme Scientifics LLC., PA, USA). The glycosidase-treated or untreated proteins were mixed with PBS, HIDROX, or HT. The final concentration of HIDROX and HT in the mixture was 0.90 mg/mL. These mixtures were incubated at 25 °C for 0 (no reaction time) or 24 h and then mixed with sodium dodecyl sulfate (SDS)-buffer with 2-mercaptoethanol (FUJIFILM Wako Pure Chemical Co.). The samples were subjected to SDS–polyacrylamide gel electrophoresis (SDS–PAGE) and Western blotting as described in [11]. S protein S1 subunit, S protein RBD, S protein S2 subunit, and N protein were detected using primary and secondary antibodies as shown in Appendix A.

### 2.5. Real-Time RT-PCR

SARS-CoV-2 solution (7.25 log_10_ TCID_50_/mL) was mixed with nine volumes of HIDROX or HT (final concentration was 0.90 mg/mL). As a diluent control, PBS was mixed with the viral suspension. The mixtures were placed at 25 °C for 0 (no reaction time) or 24 h, and then RNA was extracted using ISOGEN-LS (Nippon Gene, Tokyo, Japan) according to the manufacturer’s protocol. A total of 500 ng of RNA thus obtained was reverse transcribed using FastGene cDNA Synthesis 5× ReadyMix OdT (NIPPON Genetics Co, Ltd., Tokyo, Japan). Then real-time RT-PCR was performed using EagleTaq Master Mix with ROX (F. Hoffmann-La Roche Ltd., Basel, Switzerland) with the following primers and probe; NIID_2019-nCoV_N_F2 (AAATTTTGGGGACCAGGAAC), NIID_2019-nCoV_N_R2 (TGGCAGCTGTGTAGGTCAAC), NIID_2019-nCoV_N_P2 (FAM-ATGTCGCGCATTGGCATGGA-TAMRA) [27]. The real-time RT-PCR conditions were as follows: 50 °C for 2 min; 95 °C for 10 min; and 45 cycles of 95 °C for 15 s and 60 °C for 1 min.

### 2.6. Evaluation of Virucidal Activity of HIDROX-Containing Cream against SARS-CoV-2

A total of 20 mg of test cream was applied on 2.25 cm^2^ (1.5 cm × 1.5 cm) of polyethylene terephthalate film (AS ONE Co., Ltd., Osaka, Japan). The lid of a 12-well plate (Nunc, Rochester, NY, USA) was turned over and 5.25 log_10_ TCID_50_/60 µL of SARS-CoV-2 solution was placed on the inside of the lid. The viral suspension was covered by the cream-coated film such that the cream contacted the viral suspension. The 12-well plate was incubated between 10 min and 6 h at 25 °C. Subsequently, the viral suspension was recovered and inoculated into cells; a ten-fold serial dilution was performed. After incubation for 1 h at 37 °C, the virus-containing cell culture medium was removed and new VGM was added. After incubation for 3 d at 37 °C, the viral titer (log_10_ TCID_50_/mL) was calculated. Since none of the test samples showed cytotoxicity, the detection limit of the viral titer was set to 1.25 log_10_ TCID_50_/mL in all the test groups.

### 2.7. Statistical Analysis

Student’s *t*-test was performed to determine statistically significant differences between the control group and each test sample group. *P* values of less than 0.05 were considered statistically significant.

## 3. Results

### 3.1. Evaluation of Time- and Concentration-Dependent Virucidal Activity of HIDROX against SARS-CoV-2

First, the virucidal activity of HIDROX against SARS-CoV-2 was evaluated. HIDROX inactivated SARS-CoV-2 in a time- and concentration-dependent manner. HIDROX (4.50 mg/mL) inactivated 99.68% virus (2.50 log_10_ TCID_50_/mL reduction of viral titer when compared to the PBS group in the same reaction time) in 0.5 h. Moreover, 0.90 and 0.45 mg/mL HIDROX inactivated 98.53% and 90.00% virus (1.83 and 1.00 log_10_ TCID_50_/mL reduction, respectively) in 1 h (Figure 1A). In addition, 5.63 and 11.25 mg/mL HIDROX inactivated 86.67% and 95.78% virus (0.88 and 1.38 log_10_ TCID_50_/mL reduction, respectively) in 5 min (Figure 1B).

### 3.2. Comparison of Virucidal Activity of HIDROX and HT against SARS-CoV-2

The virucidal activities of HIDROX and HT against SARS-CoV-2 were compared. In addition to 0.90 mg/mL HIDROX and HT, 0.05 mg/mL HT, which is the concentration of HT in 0.90 mg/mL HIDROX, was tested. At 3 h, 0.90 mg/mL HIDROX exhibited significant virucidal activity, whereas 0.90 and 0.05 mg/mL HT did not. Although 0.90 mg/mL of both HIDROX and HT inactivated ≥99.98% of virus (≥3.63 log_10_ TCID_50_/mL reduction when compared to the PBS group in the same reaction time; the viral titer was below the detection limit) in 24 h, the virucidal activity of 0.05 mg/mL of HT was limited (0.63 log_10_ TCID_50_/mL reduction) (Figure 2). Thus, the SARS-CoV-2-inactivating activity of HIDROX is more potent than that of pure HT.

### 3.3. Evaluating the Impact of HIDROX and HT on SARS-CoV-2 Structural Proteins

To evaluate the impact of HIDROX and HT on SARS-CoV-2 structural proteins, S protein S1 and S2 subunits and N protein treated with PBS (diluent control), HIDROX, and HT were analyzed by Western blotting. The results of Western blotting to detect S protein S1 and S2 subunits on PBS-treated-viruses revealed two bands of ~250 and ~100 kDa at both 0 h (no reaction time) and 24 h reaction time. The higher molecular weight band was presumably full-length S protein and the lower molecular weight band was the cleaved S protein, corresponding to the S1 or S2 subunit. HIDROX- and HT-treated viruses also exhibited these two bands at 0 h. On the other hand, in the detection of S1 subunit at 24 h, the intensity of these two bands was weaker in HIDROX/HT-treated viruses than that in PBS-treated viruses. In addition, the other ladder with >250 kDa appeared in HIDROX/HT-treated viruses (Figure 3A left panel). In the detection of S2 subunit at 24 h, the intensity of the band of approximately 100 kDa size was slightly weaker in HIDROX-treated viruses than in PBS-treated viruses, and the ladder with >250 kDa size was detected in HIDROX/HT-treated viruses (Figure 3A middle panel). Detection of the N protein by Western blotting revealed no difference in the intensity of specific bands among PBS-, HIDROX-, and HT-treated viruses at both 0 and 24 h (Figure 3A right panel). Next, to evaluate the impact of HIDROX and HT on RBD (located on the S1 subunit of S protein), Western blotting was also performed. Although a main band of ~65 kDa was observed in PBS-, HIDROX, and HT-treated recombinant proteins at 0 h, the band almost disappeared and a strong band of >150 kDa appeared after HIDROX treatment at 24 h. By contrast, both ~65 and >150 kDa bands were detectable after HT treatment for 24 h (Figure 3B left panel). Although the N protein is visible in the blot (Figure 3A), it is possible that HIDROX and HT did not contact the N protein located inside the virus particles. To evaluate the direct impact of HIDROX and HT on N protein, the recombinant N protein was mixed with PBS, HIDROX, or HT. The intensity of the ~50 kDa band was slightly weaker in HIDROX-treated protein than that in PBS-treated protein at 24 h reaction time; however, the decrease in intensity was not observed in HT-treated protein. In addition, several bands of >100 kDa appeared under both HIDROX- and HT treatment (Figure 3B right panel). Thus, HIDROX and HT induce changes that affect the molecular weight of viral structural proteins, especially the S protein expressed on the surface of viral particles. Moreover, HIDROX is more potent than HT.

### 3.4. Evaluation of the Interaction of HIDROX or HT with Carbohydrate Chains Expressed on S Proteins

Because the S protein of SARS-CoV-2 is highly glycosylated due to post-translational modifications [28], the interaction of HIDROX and HT with carbohydrate chains attached to S protein was evaluated. Recombinant S protein S1 subunit, RBD, and S2 subunit, untreated (glycosylated) or pretreated with glycosidase (deglycosylated), were mixed with PBS, HIDROX, or HT. After 24 h reaction time, Western blotting was performed. For the S1 subunit, ~150 and ~75 kDa bands were detected in glycosylated- and deglycosylated PBS-treated proteins, respectively. However, these two bands weakened after HIDROX treatment. In addition, bands/ladders >250 kDa appeared after HIDROX and HT treatment in both glycosylated and deglycosylated proteins (Figure 4 left panel). For RBD, the molecular weight of the specific band observed in PBS-treated protein was slightly reduced due to deglycosylation. Both the glycosylated and deglycosylated bands almost disappeared after HIDROX treatment; bands/ladders >100 kDa appeared after both HIDROX and HT treatments (Figure 4 middle panel). For the S2 subunit, the molecular weight of the main band observed in PBS-treated protein was slightly reduced due to deglycosylation. In addition, weak bands >~250 kDa were observed in PBS-treated protein. However, both glycosylated and deglycosylated main bands disappeared after HIDROX treatment. By contrast, the bands or ladder with >~250 kDa remained (or appeared) in HIDROX- and HT-treated proteins (Figure 4 right panel). Thus, the carbohydrate chains on S protein were not the site of action of HIDROX and HT.

### 3.5. Evaluating the Impact of HIDROX and HT on SARS-CoV-2 Genome

To evaluate the impact of HIDROX and HT on SARS-CoV-2 genome, the viral RNA was extracted from PBS-, HIDROX-, and HT-treated viruses and real-time RT-PCR targeting the region with approximately 150 bp located on the N gene [27] was performed. Although the Ct values were similar among all treatment groups at 0 h (no reaction time), at 24 h, the Ct values were 3.19- and 2.63-fold higher in the HIDROX and HT groups than that in the PBS group, respectively (Figure 5). Although the real-time RT-PCR analysis in this study was not the absolute quantification assay to validate the copy number of viral RNAs, the increase in the Ct value seemed to be correlated with the decrease in the amount of targeted viral RNA region. Hence, this result indicates that HIDROX and HT disrupt the viral genome.

### 3.6. Evaluating the Time- and Concentration-Dependent Virucidal Activity of HIDROX-Containing Cream against SARS-CoV-2

Finally, the virucidal activity of the HIDROX-containing cream, which can be used for topical application, such as a hand cream, against SARS-CoV-2 was evaluated. Twenty mg cream containing HIDROX, applied on 2.25 cm^2^ of film, exhibited time- and concentration-dependent SARS-CoV-2-inactivating activity. The cream containing 10% and 5% HIDROX inactivated 94.38% and 79.47% virus (1.25 and 0.69 log_10_ TCID_50_/mL reduction of viral titer when compared to the 0% HIDROX group in the same reaction time, respectively) in 10 min. Moreover, the cream containing 2% HIDROX inactivated 94.38% virus (1.25 log_10_ TCID_50_/mL reduction of viral titer) in 30 min (Figure 6).

## 4. Discussion

In this study, the SARS-CoV-2 inactivating activities of HIDROX and HT, especially the superiority of HIDROX to pure HT, was shown in vitro (Figure 1 and Figure 2). We previously showed that HT inactivated enveloped viruses, namely, IAV and Newcastle disease virus, and not non-enveloped viruses, namely, bovine rotavirus and fowl adenovirus [17]. In this report, the hemagglutination activity of HA protein and neuraminidase activity of NA protein, expressed on the surface of IAV, seemed unaffected by HT, and the impact of HT on viral structural proteins was unclear. By contrast, other studies have reported that HT binds to the hydrophobic cavity located on the transmembrane subunit gp41 of HIV-1 envelope glycoprotein, which is important for fusion of viral and host cellular membranes [29,30]. In addition, HT binds to the integrase active site of HIV-1 integrase, which inhibits integrase activity [31]. In this study, HIDROX and HT seemed to induce structural changes in the S protein, which is expressed on the surface of SARS-CoV-2, regardless of glycosylation. Brudzynski et al. mentioned in their review article that a polyphenol interacts with a protein via hydrogen and hydrophobic bonds, which induces the aggregation of the polyphenol-protein complex [32]. It is hence possible that the aggregation of S proteins occurred by HIDROX/HT treatment and resulted in the appearance of the bands/ladders with a high molecular weight in Western blotting. In the detection of S1 subunit, HIDROX-treated virus and HIDROX-treated recombinant protein showed almost the same band change pattern at 24 h; namely, a decrease in the main band compared to that in the PBS-treated group as well as the appearance of the ladder of >250 kDa (Figure 3A left panel, Figure 4 left panel). On the other hand, in the detection of S2 subunit, a different band change pattern was noted between the HIDROX-treated virus and the HIDROX-treated recombinant protein at 24 h; for example, while the decrease in the main band was small in HIDROX-treated virus, this decrease was potent in HIDROX-treated recombinant protein (Figure 3A middle panel, Figure 4 right panel). The S2 subunit is covered by the S1 subunit, and some regions of the S2 subunit are inside the viral particle. Such structural features of the S2 subunit expressed on the viral particle may make contact with HIDROX difficult, which could have contributed to the smaller effect by HIDROX in the virus when compared to that in the recombinant protein. On the other hand, the impact on N protein seemed to be lower than that on S proteins (Figure 3 and Figure 4). Thus, whereas HT can interact with several viral proteins, some distinctive amino acid motif(s) with high binding affinity to HT may be present. To identify the binding sites of HT to SARS-CoV-2 structural proteins, computational docking analysis will be useful in the future. Our previous study revealed the possibility that the viral envelope was the site of action of HT because non-enveloped viruses are unaffected by HT [17]. Moreover, HIDROX or HT treatment disrupts the SARS-CoV-2 genome (Figure 5). This result suggests that the viral particle becomes fragile due to HIDROX or HT treatment, and then HIDROX or HT directly contact the viral genome located inside the viral particle or the viral genome leaked out. Although no direct evidence has been shown, it is possible that HIDROX or HT impact the viral envelope and contribute to the fragility of SARS-CoV-2 particle. Regarding the mechanism of the viral genome disruption, Furukawa et al. demonstrated that green tea catechins generated reactive oxygen in the presence of metal ions, which result in DNA damage [33]. Such nucleotide damage mediated by reactive oxygen could also have been induced by HIDROX/HT treatment.

Similar to previous reports showing the beneficial bioactivities of HIDROX and HT [20,21,22], HIDROX showed stronger SARS-CoV-2-inactivating activity than pure HT. Studies on the antiviral activities of polyphenols [14] have suggested the presence of additive or synergistic antiviral effect by combination of different phenolic compounds [34,35]. In addition to HT, HIDROX contains a various phenolic compounds, including oleuropein, which is a precursor of HT and is an antiviral [15,36]. Because of the disruptions of multiple viral structures by the compounds with different modes of action, or the enhancement of the impact on one viral structure by the compounds with analogous mode of actions, HIDROX may have more potent virucidal activity than individual compounds. Although this study showed the superiority of HIDROX in inactivating SARS-CoV-2 in vitro, in the future, in vivo studies using animal models should be performed to evaluate the effectiveness of HIDROX as an antiviral. The requirement of high HIDROX concentrations for rapid and potent virucidal activity in vitro, and the fast metabolism or excretion of HT and other olive-derived phenolic compounds [15] are concerns for accomplishing the protective effect of HIDROX. However, oral and intraperitoneal administration of polyphenol-rich plant extracts and phenolic compounds had a protective effect in IAV-infected mice [37,38]. It was suggested that not only direct antiviral activity of the phenolic but also its anti-inflammatory activity contributed to the protective effect [38]. Because excessive inflammation is associated with the exacerbation of COVID-19 symptoms [39], the intake of HIDROX may protect against COVID-19.

Here we also showed the virucidal activity of HIDROX-containing creams (Figure 6). In general, poor hand hygiene is considered to increase a risk of infection and hand washing or use of antiviral disinfectants may reduce the risk [40,41]. However, the frequent use of hand hygiene agents causes skin irritation in healthcare workers. Thus, hand lotions and creams can improve the skin condition [42]. In this study, 20 mg of HIDROX-containing cream coated on 2.25 cm^2^ of film significantly inactivated SARS-CoV-2, with a viral titer of 5.25 log_10_ TCID_50_, within 10 or 30 min in a concentration-dependent manner. Because this viral titer is assumed to be considerably higher than the amount of viruses remaining in the contaminated environment [43], the protective effect against SARS-CoV-2 infection is expected when the HIDROX-containing cream is used topically.

Our findings reveal the possibility that HT-rich olive extract can be safely applied to protect against various viruses and aid in controlling SARS-CoV-2.

## Figures and Tables

**Figure 1 viruses-13-00232-f001:**
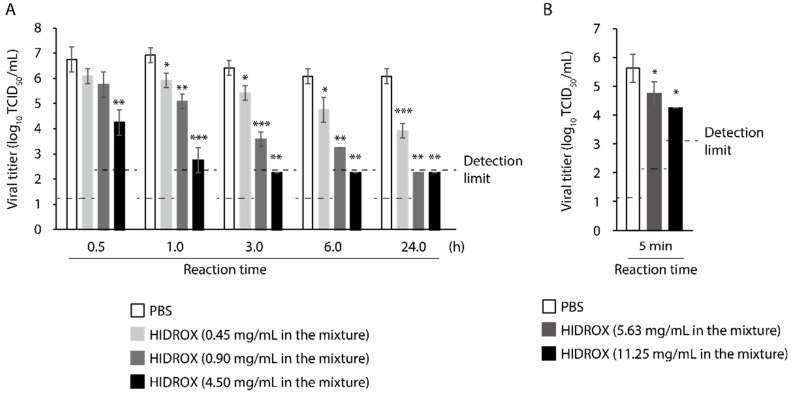
Time- and concentration-dependent virucidal activity of hydroxytyrosol (HT)-rich aqueous olive pulp extract (HIDROX) against severe acute respiratory syndrome coronavirus 2 (SARS-CoV-2). (**A**,**B**) SARS-CoV-2 was mixed with HIDROX. The final concentrations of HIDROX in the mixture were 0.45, 0.90, and 4.50 mg/mL (**A**) or 5.63 and 11.25 mg/mL (**B**). As a diluent control, PBS was mixed with the viral suspension. The mixtures were incubated at 25 °C for 0.5–24 h (**A**) or 5 min (**B**) and then the viral titers were evaluated. The detection limits of viral titer are 1.25 log_10_ 50% tissue culture infective dose (TCID_50_)/mL in the PBS and HIDROX (0.45 mg/mL) groups, 2.25 log_10_ TCID_50_/mL in the HIDROX (0.9, 4.50, and 5.63 mg/mL) groups, and 3.25 log_10_ TCID_50_/mL in the HIDROX (11.25 mg/mL) group. Results are indicated as mean ± SD (*n* = 3–4 per group). Student’s *t*-test was performed to evaluate the statistically significant difference between the PBS and HIDROX groups; * *p* < 0.05; ** *p* < 0.01; *** *p* < 0.001.

**Figure 2 viruses-13-00232-f002:**
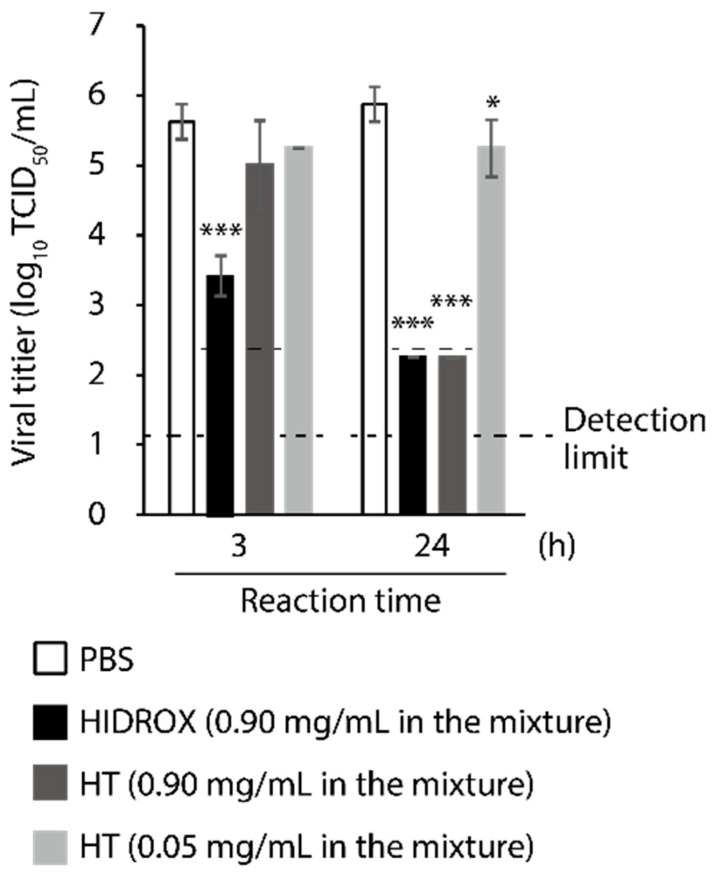
Comparison of virucidal activity of hydroxytyrosol (HT)-rich aqueous olive pulp extract (HIDROX) and HT against severe acute respiratory syndrome coronavirus 2 (SARS-CoV-2). SARS-CoV-2 was mixed with HIDROX or HT. The final concentrations of HIDROX and HT in the mixture were 0.90 mg/mL, and 0.05 and 0.90 mg/mL, respectively. PBS was mixed with the viral suspension as the diluent control. The mixtures were incubated at 25 °C for 3 or 24 h, and then the viral titers were evaluated. The detection limits of viral titer are 1.25 log_10_ 50% tissue culture infective dose (TCID_50_)/mL in the PBS and HT (0.05 mg/mL) groups and 2.25 log_10_ TCID_50_/mL in the HIDROX (0.9 mg/mL) and HT (0.90 mg/mL) groups. Results are indicated as mean ± SD (*n* = 4 per group). Student’s *t*-test was performed to evaluate the statistically significant difference between the PBS group and each test group; * *p* < 0.05; *** *p* < 0.001.

**Figure 3 viruses-13-00232-f003:**
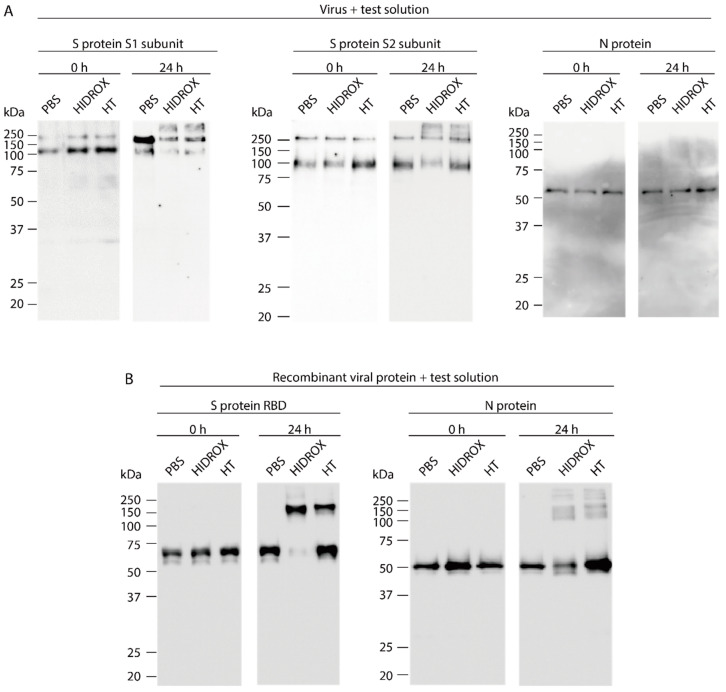
Impact of hydroxytyrosol (HT)-rich aqueous olive pulp extract (HIDROX) and HT on severe acute respiratory syndrome coronavirus 2 (SARS-CoV-2) proteins. (**A**) SARS-CoV-2 was mixed with HIDROX or HT. The final concentrations of HIDROX and HT in the mixture were 0.90 mg/mL. As a diluent control, PBS was mixed with the viral suspension. The mixtures were incubated at 25 °C for 0 (no reaction time) or 24 h, and then Western blotting was performed. The images on the left, middle, and right show the results of Western blotting to detect S protein S1 subunit, S protein S2 subunit, and N protein, respectively. (**B**) The recombinant S protein RBD or N protein was mixed with PBS, HIDROX, or HT. The final concentration of HIDROX and HT in the mixture was 0.90 mg/mL. The final concentration of recombinant viral proteins in the mixture was approximately 10 µg/mL. The mixtures were incubated at 25 °C for 0 (no reaction time) or 24 h and then Western blotting was performed. The images on the left and right show the results of Western blotting to detect S protein RBD and N protein, respectively.

**Figure 4 viruses-13-00232-f004:**
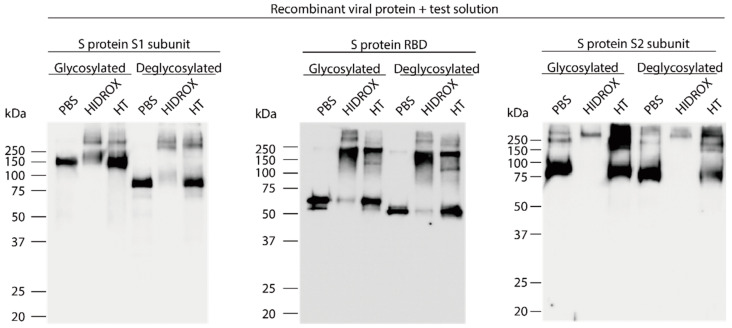
Impact of hydroxytyrosol (HT)-rich aqueous olive pulp extract (HIDROX) or HT on carbohydrate chains expressed on the S protein. The glycosylated or deglycosylated recombinant S protein S1 subunit, RBD, and S2 subunit were mixed with PBS, HIDROX, or HT. The final concentration of HIDROX and HT in the mixture was 0.90 mg/mL. The final concentration of recombinant S proteins in the mixture was approximately 10 µg/mL. The mixtures were incubated at 25 °C for 24 h and then Western blotting was performed. The images on the left, middle, and right show the S1 subunit, RBD, and S2 subunit, respectively.

**Figure 5 viruses-13-00232-f005:**
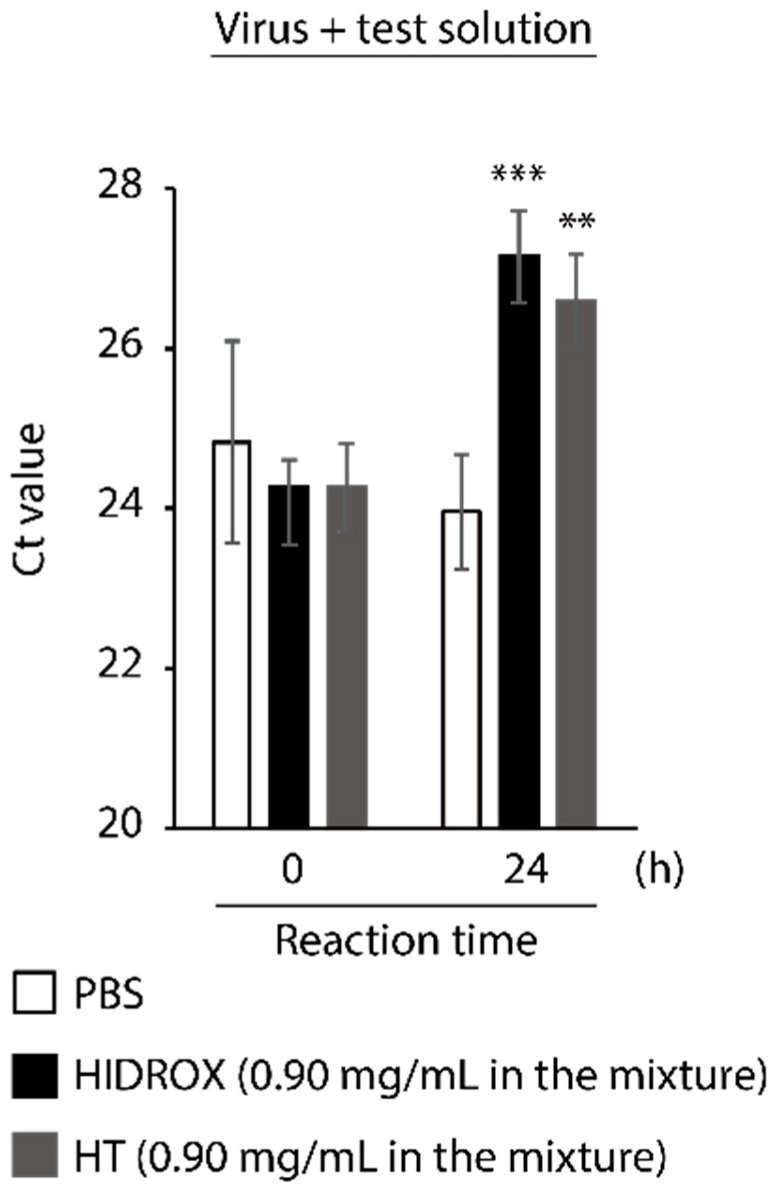
Impact of hydroxytyrosol (HT)-rich aqueous olive pulp extract (HIDROX) and HT on severe acute respiratory syndrome coronavirus 2 (SARS-CoV-2) genome. SARS-CoV-2 was mixed with HIDROX or HT. The final concentrations of HIDROX and HT in the mixture were 0.90 mg/mL. As a diluent control, PBS was mixed with the viral suspension. The mixtures were incubated at 25 °C for 0 (no reaction time) or 24 h. Then real-time RT-PCR was performed and Ct value was evaluated. Results are indicated as mean ± SD (*n* = 4 per group). Student’s *t*-test was performed to evaluate the statistically significant difference between the PBS group and each test solution group; ** *p* < 0.01; *** *p* < 0.001.

**Figure 6 viruses-13-00232-f006:**
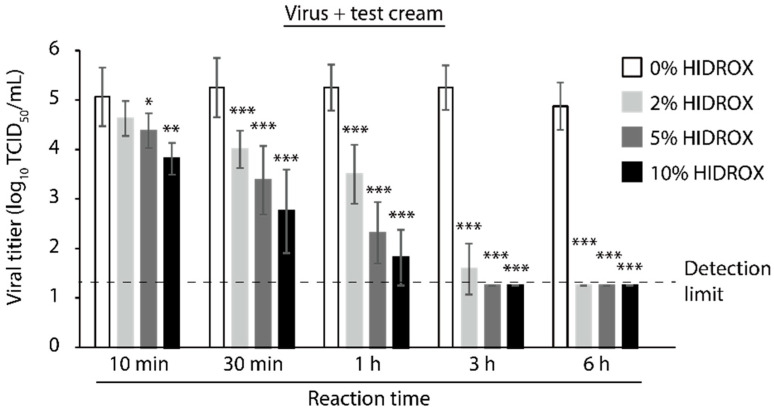
Time- and concentration-dependent virucidal activity of hydroxytyrosol (HT)-rich aqueous olive pulp extract (HIDROX)-containing cream against severe acute respiratory syndrome coronavirus 2 (SARS-CoV-2). SARS-CoV-2 was covered by 0%, 2%, 5%, and 10% HIDROX-containing cream-coated films and incubated for 10 min–6 h at 25 °C. After the predetermined reaction times, the viral solutions were recovered and the viral titers were calculated. The detection limit of the viral titer is 1.25 log_10_ 50% tissue culture infective dose (TCID_50_)/mL in all groups. Results are indicated as mean ± SD (*n* = 4–8 per group). Student’s *t*-test was performed to evaluate the statistically significant difference between the 0% HIDROX group and other groups; * *p* < 0.05; ** *p* < 0.01; *** *p* < 0.001.

## Data Availability

The data presented in this study are available in Appendix A.

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
