# Peer review of "The SARS-CoV-2-Inactivating Activity of Hydroxytyrosol-Rich Aqueous Olive Pulp Extract (HIDROX®) and Its Use as a Virucidal Cream for Topical Application"

_viruses, 2021, doi:10.3390/v13020232_

Round 1

Reviewer 1 Report

This is a very good, well controlled study demonstrating the effectiveness of HIDROX against SARS-CoV-2. 

Can the authors give more detail in the conclusions as to why they think the protein >150KdA became larger with HIDROX treatment (Line 214).

Line 276 - the authors state 'This result indicates that HIDROX and HT disrupt the viral genome.' The up to 10 fold reduction in viral quantification is quite weak compared to the reduction in virus titre by TCID50. Please can the authors add further comments to the conclusions section to account for this. Also, can they suggest a mechanism for how the viral genome is disrupted.

Section 3.6 Line 294 & 296. Please check the values of 94.38% virus inactivation for 10% at 10 minutes and 2% at 30 minutes. These values are identical, is that correct? The values appear different on the bar charts, however the starting titre of the 0% HIDROX appears slightly higher for 30 min, which may account for this?

For future studies it would be interesting to know if the activity of HIDROX was maintained over time e.g. if HIDROX was applied to a surface how long before it lost its virucidal activity?

Author Response

We are grateful to the reviewers for their valuable comments and helpful suggestions. We have revised the manuscript based on the reviewer’s comments. All the changes in the revised manuscript are highlighted in red color font. Further, we have provided our point-by-point responses to the reviewer’s comments below:

Response to Reviewer 1 Comments

Point 1: Can the authors give more detail in the conclusions as to why they think the protein >150KdA became larger with HIDROX treatment (Line 214).

Response 1: We have added the details related to the appearance of high molecular band/ladder by HIDROX/HT treatment in the Discussion section of the revised manuscript.

Point 2: Line 276 - the authors state 'This result indicates that HIDROX and HT disrupt the viral genome.' The up to 10 fold reduction in viral quantification is quite weak compared to the reduction in virus titre by TCID50. Please can the authors add further comments to the conclusions section to account for this. Also, can they suggest a mechanism for how the viral genome is disrupted.

Response 2: We highly appreciate the reviewer’s valuable comments. We believe that it is difficult to compare the results of real-time RT-PCR with those of viral titer because our real-time RT-PCR only targeted the region with approximately 150 bp located on the N gene, and we did not analyze the entire region of full viral genome. Hence, we could not estimate the degree of overall disruption in the viral genome. We have revised the description related to this point in order to avoid misleading the extended interpretation in Section 3.5. In addition, we have added the discussion related to point on the mechanism of HIDROX/HT-induced viral genome disruption in the Discussion section of the revised manuscript.

Point 3: Section 3.6 Line 294 & 296. Please check the values of 94.38% virus inactivation for 10% at 10 minutes and 2% at 30 minutes. These values are identical, is that correct? The values appear different on the bar charts, however the starting titre of the 0% HIDROX appears slightly higher for 30 min, which may account for this?

Response 3: We apologize for the incomprehensible description. As the reviewer assumed, we did calculate the percent of virus inactivation according to the differences in the viral titers between the HIDROX-free control group and each of the test group in a same reaction time. We have added additional explanations to the relevant portions of the revised manuscript in Sections 3.1., 3.2., and 3.6.

Point 4: For future studies it would be interesting to know if the activity of HIDROX was maintained over time e.g. if HIDROX was applied to a surface how long before it lost its virucidal activity?

Response 4: We are grateful to the reviewer for their valuable suggestion. We are interested in collecting more practical data, including those on the duration of the activity and stability in the future study. As a suggestive result, while the viral titers of 0% HIDROX group in 3 h and 0% HIDROX group in 1 h were similar, the viral titers of 2% HIDROX group in 3 h were lower than those of 2% HIDROX group in 1 h (Figure 6). This result suggests that the virucidal activity of HIDROX applied to a surface was maintained for >1 h. In the future, we hope to evaluate the duration of this activity over a longer duration.

Reviewer 2 Report

The present work entitled "The SARS-CoV-2-inactivating activity of hydroxytyrosol-rich aqueous olive pulp extract (HIDROX®) and its use as a virucidal cream for topical application" is well argued and presented, with a well-described and very detailed methodology. It is worth highlighting the objective of this work, the search of antiviral compounds to combat SARS-CoV-2, remarkably when it is based on natural candidates. The discussion is well conducted, especially when addressing the high requirement of HIDROX concentrations for in vitro activity from the point of view of translating it to the in vivo model (where higher concentrations of antiviral compounds are usually required). The final conclusion obtained is interesting and with future perspectives, highlighting its use in cream format.

Points to improve:

  • Table S1 (mentioned on page 3, lines 114 and 128) is absent. Please include it with the manuscript.
  • The detection limits of viral titer are different depending on the concentration of the compound used (Figures 1 and 2). Clarify why it is so.
  • When evaluating the impact of compounds on SARS-CoV-2 structural proteins:
  1. If possible, it is preferable to normalize the amount of protein used in Western Blot assays including the detection of an irrelevant (non-structural) protein. The possibility of seeing the same intensity of the unaffected bands would help make the impact of the compounds on the target protein more visible. If not possible, at least the quality of the Western Blots should be improved.
  2. The results obtained when using whole virus vs. recombinant protein are different (for example, in figure 4, first panel, S protein S1 subunit glycosylated, there is no apparent effect of HIDROX and HT in contrast to what is observed in fig 3A, first panel (the intensity of these bands weaken), both at 24 h. The same is observed in the S2 subunit with HT. Please include a possible explanation in the discussion.
  • When evaluating the impact of HIDROX and HT on SARS-CoV-2 genome, the Ct values are correlated with the decrease of the amount of viral RNA (page 7, lanes 273-275) but there is no standard curve validating the number of copies obtained by transcription, please clarify it.

Minor points:

  • Page 1, lines 32-33: What was declared a pandemic (as mentioned in ref. 2) is not SARS-CoV-2 but the disease caused by the virus, named COVID-19.
  • Page 1, line 34: State what the abbreviation COVID-19 is equivalent to (the name of the disease).
  • Page 5, line 211-212: Since a Western Blot was previously performed, add the word “also” (Next, to evaluate the impact of HIDROX and HT on RBD (located on the S1 subunit of S protein), western blotting was also)

Author Response

We are grateful to the reviewers for their valuable comments and helpful suggestions. We have revised the manuscript based on the reviewer’s comments. All the changes in the revised manuscript are highlighted in red color font. Further, we have provided our point-by-point responses to the reviewer’s comments below:

Response to Reviewer 2 Comments

Point 1: Table S1 (mentioned on page 3, lines 114 and 128) is absent. Please include it with the manuscript.

Response 1: We apologize for having missed including Table S1 in the manuscript. We have added Table S1 to the last page of the revised manuscript.

Point 2: The detection limits of viral titer are different depending on the concentration of the compound used (Figures 1 and 2). Clarify why it is so.

Response 2: We have added more detailed explanation about setting of the detection limit in Sections 2.3. and 2.6., as advised.

Point 3: When evaluating the impact of compounds on SARS-CoV-2 structural proteins:

  1. If possible, it is preferable to normalize the amount of protein used in Western Blot assays including the detection of an irrelevant (non-structural) protein. The possibility of seeing the same intensity of the unaffected bands would help make the impact of the compounds on the target protein more visible. If not possible, at least the quality of the Western Blots should be improved.

Response 3: We agree with the reviewer’s comment, but could not normalize the amount of protein in the western blotting assay. Therefore, to obtain more accurate results, the same experiments were repeated more than twice to confirm the reproducibility of the results. We attempted to improve the quality of the results and have changed some pictures to the new pictures (Specifically, the pictures of Fig 3A_S1 subunit_24 h, Fig 3A_S2 subunit_0 h and 24 h, Fig 4_S1 subunit, and Fig 4_RBD were replaced). Accordingly, some descriptions in Sections 3.3. and 3.4. have been revised to make the results more accurate and comprehensive.

Point 4:

  1. The results obtained when using whole virus vs. recombinant protein are different (for example, in figure 4, first panel, S protein S1 subunit glycosylated, there is no apparent effect of HIDROX and HT in contrast to what is observed in fig 3A, first panel (the intensity of these bands weaken), both at 24 h. The same is observed in the S2 subunit with HT. Please include a possible explanation in the discussion.

Response 4: As mentioned earlier, we repeated the experiments and acquired more probable outputs than before. As a result, we observed the same band change patterns of S1 subunit between HIDROX-treated virus and HIDROX-treated recombinant protein at 24 h. On the other hand, the band change patterns of the S2 subunit between HIDROX-treated virus and HIDROX-treated recombinant protein at 24 h remained slightly different. Hence, we added the discussion about these discrepancies in the Discussion section of the revised manuscript.

Point 5:

When evaluating the impact of HIDROX and HT on SARS-CoV-2 genome, the Ct values are correlated with the decrease of the amount of viral RNA (page 7, lanes 273-275) but there is no standard curve validating the number of copies obtained by transcription, please clarify it.

Response 5: We appreciate the reviewer’s comments and have revised the description to clarify the same in Section 3.5. of the revised manuscript.

Point 6:

Minor points:

Page 1, lines 32-33: What was declared a pandemic (as mentioned in ref. 2) is not SARS-CoV-2 but the disease caused by the virus, named COVID-19.

Page 1, line 34: State what the abbreviation COVID-19 is equivalent to (the name of the disease).

Page 5, line 211-212: Since a Western Blot was previously performed, add the word “also” (Next, to evaluate the impact of HIDROX and HT on RBD (located on the S1 subunit of S protein), western blotting was also)

Response 6: We appreciate the reviewer’s comments. We have revised the manuscript in accordance with the same.

Round 2

Reviewer 2 Report

A very good revision was made. Excellent!

There is only a minor point in the page 3, "2.3. Evaluation of virucidal activity of HIDROX and HT solutions against SARS-CoV-2" section, remove the comma in the phrase "A detection limit was set higher, in the group treated with a solution of higher cytotoxicity."

Good luck!